# Elementary Estimators for Graphical Models

**Eunho Yang**
IBM T.J. Watson Research Center
eunhyang@us.ibm.com

**Aurélie C. Lozano**
IBM T.J. Watson Research Center
aclozano@us.ibm.com

**Pradeep Ravikumar**
University of Texas at Austin
pradeepr@cs.utexas.edu

## Abstract

We propose a class of closed-form estimators for sparsity-structured graphical models, expressed as exponential family distributions, under high-dimensional settings. Our approach builds on observing the precise manner in which the classical graphical model MLE "breaks down" under high-dimensional settings. Our estimator uses a carefully constructed, well-defined and closed-form backward map, and then performs thresholding operations to ensure the desired sparsity structure. We provide a rigorous statistical analysis that shows that surprisingly our simple class of estimators recovers the same asymptotic convergence rates as those of the $\ell_1$-regularized MLEs that are much more difficult to compute. We corroborate this statistical performance, as well as significant computational advantages via simulations of both discrete and Gaussian graphical models.

## 1 Introduction

Undirected graphical models, also known as Markov random fields (MRFs), are a powerful class of statistical models, that represent distributions over a large number of variables using graphs, and where the structure of the graph encodes Markov conditional independence assumptions among the variables. MRFs are widely used in a variety of domains, including natural language processing [1], image processing [2, 3, 4], statistical physics [5], and spatial statistics [6]. Popular instances of this class of models include Gaussian graphical models (GMRFs) [7, 8, 9, 10], used for modeling continuous real-valued data, and discrete graphical models including the Ising model where each variable takes values in a discrete set [10, 11, 12]. In this paper, we consider the problem of high-dimensional estimation, where the number of variables $p$ may exceed the number of observations $n$. In such high-dimensional settings, it is still possible to perform consistent estimation by leveraging low-dimensional structure. Sparse and group-sparse structural constraints, where few parameters (or parameter groups) are non-zero, are particularly pertinent in the context of MRFs as they translate into graphs with few edges.

A key class of estimators for learning graphical models has thus been based on maximum likelihood estimators (MLE) with sparsity-encouraging regularization. For the task of sparse GMRF estimation, the state-of-the-art estimator minimizes the Gaussian negative log-likelihood regularized by the $\ell_1$ norm of the entries (or the off-diagonal entries) of the concentration matrix (see [8, 9, 10]). Strong statistical guarantees for this estimator have been established (see [13] and references therein). The resulting optimization problem is a log-determinant program, which can be solved in polynomial time with interior point methods [14], or by co-ordinate descent algorithms [9, 10]. In a computationally simpler approach for sparse GMRFs, [7] proposed the use of neighborhood selection, which consists of estimating conditional independence relationships separately for each node in the graph, via $\ell_1$-regularized linear regression, or LASSO [15]. They showed that the procedure can

consistently recover the sparse GMRF structure even under high-dimensional settings. The neighborhood selection approach has also been successfully applied to discrete Markov random fields. In particular, for binary graphical models, [11] showed that consistent neighborhood selection can be performed via $\ell_1$-regularized logistic regression. These results were generalized to general discrete graphical models (where each variable can take $m \geq 2$ values) by [12] through node-wise multi-class logistic regression with group sparsity. A related regularized convex program to solve for sparse GMRFs is the CLIME estimator [16], which reduces the estimation problem to solving linear programs. Overall, while state of the art optimization methods have been developed to solve all of these regularized (and consequently non-smooth) convex programs, their iterative approach is very expensive for large scale problems. Indeed, scaling such regularized convex programs to very large scale settings has attracted considerable recent research and attention.

In this paper, we investigate the following leading question: *"Can one devise simple estimators with closed-form solutions that are yet consistent and achieve the sharp convergence rates of the aforementioned regularized convex programs?"* This question was originally considered in the context of linear regression by [17] and to which they had given a positive answer. It is thus natural to wonder whether an affirmative response can be provided for the more complicated statistical modeling setting of MRFs as well.

Our key idea is to revisit the vanilla MLE for estimating a graphical model, and consider where it "breaks down" in the case of high-dimensions. The vanilla graphical model MLE can be expressed as a *backward mapping* [18] in an exponential family distribution that computes the model parameters corresponding to some given (sample) moments. There are however two caveats with this backward mapping: it is not available in closed form for many classes of models, and even if it were available in closed form, it need not be well-defined in high-dimensional settings (i.e. could lead to unbounded model parameter estimates). Accordingly, we consider the use of carefully constructed *proxy* backward maps that are both available in closed-form, and well-defined in high-dimensional settings. We then perform simple thresholding operations on these proxy backward maps to obtain our final estimators. Our class of algorithms is thus both computationally practical and highly scalable. We provide a unified statistical analysis of our class of algorithms for graphical models arising from general exponential families. We then instantiate our analysis for the specific cases of GMRFs and DMRFs, and show that the resulting algorithms come with strong statistical guarantees achieving near-optimal convergence rates, but doing so computationally much faster than the regularized convex programs. These surprising results are confirmed via simulation for both GMRFs and DMRFs. There has been considerable recent interest in large-scale statistical model estimation, and in particular, in scaling these to very large-scale settings. We believe our much simpler class of *closed-form* graphical model estimators have the potential to be estimators of choice in such large-scale settings, particularly if it attracts research on optimizing and scaling its closed-form operations.

## 2 Background and Problem Setup

Since most popular graphical model families can be expressed as exponential families (see [18]), we consider general exponential family distributions for a random variable $X \in \mathbb{R}^p$:

$$\mathbb{P}(X; \theta) = \exp\left\{ \langle \theta, \phi(X) \rangle - A(\theta) \right\} \tag{1}$$

where $\theta \in \Omega \subseteq \mathbb{R}^d$ is the canonical parameter to be estimated, $\phi(X)$ denotes the sufficient statistics with feature function $\phi : \mathbb{R}^p \mapsto \mathbb{R}^d$, and $A(\theta)$ is the log-partition function.

An alternative parameterization of the exponential family, to the canonical parameterization above, is via the vector of "mean parameters" $\mu(\theta) \stackrel{\text{def}}{=} \mathbb{E}_\theta[\phi(X)]$, which are the moments of the sufficient statistics $\phi(X)$ under the exponential family distribution. We denote the set of all possible moments by the moment polytope: $\mathcal{M} = \{\mu : \exists \text{ distribution } p \text{ s.t. } \mathbb{E}_p(\phi) = \mu\}$, which consist of moments of the sufficient statistics under all possible distributions. The problem of computing the mean parameters $\mu(\theta) \in \mathcal{M}$ given the canonical parameters $\theta \in \Omega$ constitutes the key machine learning problem of inference in graphical models (expressed in exponential family form (1)). Let us denote this computation via a so-called *forward mapping* $\mathcal{A} : \Omega \mapsto \mathcal{M}$. By properties of exponential family distributions, the forward mapping $\mathcal{A}$ can actually be expressed in terms of the first derivative of the log-partition function $A(\cdot)$: $\mathcal{A} : \theta \mapsto \mu = \nabla A(\theta)$. It can be shown that this map is injective (one-to-one with its range) if the exponential family is *minimal*. Moreover, it is onto the interior of

$\mathcal{M}$, denoted by $\mathcal{M}^o$. Thus, for any mean parameter $\mu \in \mathcal{M}^o$, there exists a canonical parameter $\theta(\mu) \in \Omega$ such that $\mathbb{E}_{\theta(\mu)}[\phi(X)] = \mu$. Unless the exponential family is minimal, the corresponding canonical parameter $\theta(\mu)$ however need not be unique. Thus while there will always exist a so-called *backward mapping* $\mathcal{A}^* : \mathcal{M}^o \mapsto \Omega$, that computes the canonical parameters corresponding to given moments, it need not be unique. A candidate backward map can be constructed via the conjugate of the log-partition function $A^*(\mu) = \sup_{\theta \in \Theta} \langle \theta, \mu \rangle - A(\theta)$: $\mathcal{A}^* : \mu \mapsto \theta = \nabla A^*(\mu)$.

## 2.1 High-dimensional Graphical Model Selection

We focus on the high-dimensional setting, where the number of variables $p$ may greatly exceed the sample size $n$. Under such high-dimensional settings, it is now well understood that consistent estimation is possible if structural constraints are imposed on the model parameters $\theta$. In this paper, we focus on the structural constraint of sparsity, for which the $\ell_1$ norm is known to be well-suited.

Given $n$ samples $\{X^{(i)}\}_{i=1}^n$ from $\mathbb{P}(X; \theta^*)$ that belongs to an exponential family (1), a popular class of $M$-estimators for recovering the sparse model parameter $\theta^*$ is the $\ell_1$-regularized maximum likelihood estimators:

$$\underset{\theta}{\text{minimize}} \ \langle \theta, \widehat{\phi} \rangle - A(\theta) + \lambda_n \|\theta\|_1 \tag{2}$$

where $\widehat{\phi} := \frac{1}{n} \sum_{i=1}^n \phi(X^{(i)})$ is the empirical mean of the sufficient statistics. Since the log partition function $A(\theta)$ in (1) is convex, the problem (2) is convex as well.

We now briefly review the two most popular examples of exponential families in the context of graphical models.

**Gaussian Graphical Models.** Consider a random vector $X = (X_1, \ldots, X_p)$ with associated $p$-variate Gaussian distribution $\mathcal{N}(X|\mu, \Sigma)$, mean vector $\mu$ and covariance matrix $\Sigma$. The probability density function of $X$ can be formulated as an instance of (1):

$$\mathbb{P}(X|\theta, \Theta) = \exp\left(-\frac{1}{2}\langle\!\langle \Theta, XX^\top \rangle\!\rangle + \langle \theta, X \rangle - A(\Theta, \theta)\right) \tag{3}$$

where $\langle\!\langle A, B \rangle\!\rangle$ denotes the trace inner product $\text{tr}(A B^T)$. Here, the canonical parameters are the precision matrix $\Theta$ and a vector $\theta$, with domain $\Omega := \{(\theta, \Theta) \in \mathbb{R}^p \times \mathbb{R}^{p \times p} : \Theta \succ 0, \Theta = \Theta^T\}$. The corresponding moment parameters of the graphical model distribution are given by the mean $\mu = \mathbb{E}_\theta[X]$, and the covariance matrix $\Sigma = \mathbb{E}_\theta[XX^T]$ of the Gaussian. The forward map $\mathcal{A} : (\theta, \Theta) \mapsto (\mu, \Sigma)$ computing these from the canonical parameters can be written as: $\Sigma = \Theta^{-1}$ and $\mu = \Theta^{-1}\theta$. The moment polytope can be seen to be given by $\mathcal{M} = \{(\mu, \Sigma) \in \mathbb{R}^p \times \mathbb{R}^{p \times p} : \Sigma - \mu\mu^T \succeq 0, \Sigma \succeq 0\}$, with interior $\mathcal{M}^o = \{(\mu, \Sigma) \in \mathbb{R}^p \times \mathbb{R}^{p \times p} : \Sigma - \mu\mu^T \succ 0, \Sigma \succ 0\}$. The corresponding backward map $\mathcal{A}^* : (\mu, \Sigma) \mapsto (\theta, \Theta)$ for $(\mu, \Sigma) \in \mathcal{M}^o$ can be computed as: $\Theta = \Sigma^{-1}$ and $\theta = \Sigma^{-1}\mu$.

Without loss of generality, assume that $\mu = 0$ (and hence $\theta = 0$). Then the set of non-zero entries in the precision matrix $\Theta$ corresponds to the set of edges in an associated Gaussian Markov random field (GMRF). In cases where the graph underlying the GMRF has relatively few edges, it thus makes sense to impose $\ell_1$ regularization on the off-diagonal entries of $\Theta$. Given $n$ i.i.d. random vectors $X^{(i)} \in \mathbb{R}^p$ sampled from $N(0, \Sigma^*)$, the corresponding $\ell_1$-regularized maximum likelihood estimator (MLE) is given by:

$$\underset{\Theta \succ 0}{\text{minimize}} \ \langle\!\langle \Theta, S \rangle\!\rangle - \log \det \Theta + \lambda_n \|\Theta\|_{1, \text{off}}, \tag{4}$$

where $S$ is the sample covariance matrix defined as $\sum_{i=1}^n (X^{(i)} - \overline{X})(X^{(i)} - \overline{X})^\top$, $\overline{X} := \frac{1}{n} \sum_{i=1}^n X^{(i)}$, and $\| \cdot \|_{1, \text{off}}$ is the off-diagonal element-wise $\ell_1$ norm.

**Discrete Graphical Models.** Let $X = (X_1, \ldots, X_p)$ be a random vector where each variable $X_i$ takes values in a discrete set $\mathcal{X}$ of cardinality $m$. Given a graph $G = (V, E)$, a pairwise Markov random field over $X$ is specified via nodewise functions $\theta_s : \mathcal{X} \mapsto \mathbb{R}$ for all $s \in V$, and pairwise functions $\theta_{st} : \mathcal{X} \times \mathcal{X} \mapsto \mathbb{R}$ for all $(s, t) \in E$, as

$$\mathbb{P}(X) = \exp\left\{ \textstyle\sum_{s \in V} \theta_s(X_s) + \sum_{(s,t) \in E} \theta_{st}(X_s, X_t) - A(\theta) \right\}. \tag{5}$$

This family of probability distributions can be represented using the so-called *overcomplete representations* [18] as follows. For each random variable $X_s$ and $j \in \{1, \ldots, m\}$, define nodewise

indicators $\mathcal{I}[X_s = j]$ equal to 1 if $X_s = j$ and 0 otherwise. Then pairwise MRFs in (5) can be rewritten as

$$\mathbb{P}(X) = \exp\left\{ \sum_{s \in V; j \in [m]} \theta_{s;j}\,\mathcal{I}[X_s = j] + \sum_{(s,t) \in E; j,k \in [m]} \theta_{st;jk}\,\mathcal{I}[X_s = j,\, X_t = k] - A(\theta) \right\} \quad (6)$$

for a set of parameters $\theta := \{\theta_{s;j}, \theta_{st;jk} : s,t \in V;\ (s,t) \in E;\ j,k \in [m]\}$. Given these sufficient statistics, the mean/moment parameters are given by the moments $\mu_{s;j} := \mathbb{E}_\theta\big(\mathcal{I}[X_s = j]\big) = \mathbb{P}(X_s = j; \theta)$, and $\mu_{st;jk} := \mathbb{E}_\theta\big(\mathcal{I}[X_s = j, X_t = k]\big) = \mathbb{P}(X_s = j, X_t = k; \theta)$, which precisely correspond to nodewise and pairwise marginals of the discrete graphical model. Thus, the forward mapping $\mathcal{A} : \theta \mapsto \mu$ would correspond to the inference task of computing nodewise and pairwise marginals of the discrete graphical model given the canonical parameters. A backward mapping $\mathcal{A}^* : \mu \mapsto \theta$ corresponds to computing a set of canonical parameters such that the corresponding graphical model distribution would yield the given set of nodewise and pairwise marginals. The moment polytope in this case consists of the set of all nodewise and pairwise marginals of any distribution over the random vector $X$, and hence is termed the *marginal polytope*; it is typically intractable to characterize in high-dimensions [18].

Given $n$ i.i.d. samples from an unknown distribution (6) with parameter $\theta^*$, one could consider estimating the graphical model structure with an $\ell_1$-regularized MLE: $\widehat{\theta} \in \text{minimize}_\theta -\langle \theta, \widehat{\phi} \rangle + A(\theta) + \lambda \|\theta\|_{1,E}$, where $\|\cdot\|_{1,E}$ is the $\ell_1$ norm of the edge-parameters: $\|\theta\|_{1,E} = \sum_{s \neq t} \|\theta_{st}\|$, and where we have collated the edgewise parameters $\{\theta_{st;jk}\}_{j,k=1}^m$ for an edge $(s,t) \in E$ into the vector $\theta_{st}$. However, there is an critical caveat to actually computing this regularized MLE: the computation of the log-partition function $A(\theta)$ is intractable (see [18] for details). To overcome this issue, one might consider instead the following class of $M$-estimators, discussed in [19]:

$$\widehat{\theta} \in \underset{\theta}{\text{minimize}} -\langle \theta, \widehat{\phi} \rangle + B(\theta) + \lambda \|\theta\|_{1,E}. \quad (7)$$

Here $B(\theta)$ is a variational approximation to the log-partition function $A(\theta)$ of the form: $B(\theta) = \sup_{\mu \in \mathcal{L}} \langle \theta, \mu \rangle - B^*(\mu)$, where $\mathcal{L}$ is a tractable bound on the marginal polytope $\mathcal{M}$, and $B^*(\mu)$ is a tractable approximation to the graphical model entropy $A^*(\mu)$. An example of such approximation, which we shall later leverage in this paper, is the *tree-reweighted* entropy [20] given by $B^*_{\text{trw}}(\mu) = \sum_s H_s(\mu_s) - \sum_{st} \rho_{st} I_{st}(\mu_{st})$, where $H_s(\mu_s)$ is the entropy for node $s$, $I_{st}(\mu_{st})$ is the mutual information for an edge $(s,t)$, and $\rho_{st}$ denote the edge-weights that lie in a so-called spanning tree polytope. If all $\rho_{st}$ are set to 1, this boils down to the Bethe approximation [21].

## 3 Closed-form Estimators for Graphical Models

The state-of-the-art $\ell_1$-regularized MLE estimators discussed in the previous section enjoy strong statistical guarantees but involve solving difficult non-smooth programs. Scaling them to very large-scale problems is thus an important and challenging ongoing research area.

In this paper we tackle the scalability issue at the source by departing from regularized MLE approaches and proposing instead a family of closed-form estimators for graphical models.

**Elem-GM Estimation:** $$\underset{\theta}{\text{minimize}} \|\theta\|_1 \quad (8)$$

$$\text{s.t. } \left\| \theta - \mathcal{B}^*(\widehat{\phi}) \right\|_\infty \leq \lambda_n$$

where $\mathcal{B}^*(\cdot)$ is the *proxy* of backward mapping $\mathcal{A}^*$, and $\lambda_n$ is a regularization parameter as in (2).

One of the most important properties of (8) is that the estimator is available in closed-form: $\widehat{\theta} = S_{\lambda_n}\big(\mathcal{B}^*(\widehat{\phi})\big)$, where $[S_\lambda(u)]_i = \text{sign}(u_i) \max(|u_i| - \lambda, 0)$ is the element-wise soft-thresholding function. This can be shown by the fact that the optimization problem (8) is decomposable into *independent* element-wise sub-problems, where each sub-problem corresponds to soft-thresholding.

To get some intuition on our approach, let us first revisit classical MLE estimators for graphical models as in (1), and see where they "break down" in a high-dimensional setting: $\text{minimize}_\theta \langle \theta, \widehat{\phi} \rangle - A(\theta)$. By the stationary condition of this optimization problem, the MLE estimator can be simply expressed as a backward mapping $\mathcal{A}^*(\widehat{\phi})$. There are two caveats here in high-dimensional settings.

The first is that this backward mapping need not have a simple closed-form, and is typically intractable to compute for a large number of variables $p$. The second is that the backward mapping is well-defined only for mean parameters that are in the interior $\mathcal{M}^o$ of the marginal polytope, whereas the sample moments $\widehat{\phi}$ might well lie on the boundary of the marginal polytope. We will illustrate these two caveats in the next two examples.

Our key idea is to use instead a well-defined proxy function $\mathcal{B}^*(\cdot)$ *in lieu of* the MLE backward map $\mathcal{A}^*(\cdot)$ so that $\mathcal{B}^*(\widehat{\phi})$ is both well-defined under high-dimensional settings, as well as with a simple closed-form. The optimization problem (8) seeks an estimator with minimum complexity in terms of regularizer $\| \cdot \|_1$ while being close enough to some "initial estimator" $\mathcal{B}^*(\widehat{\phi})$ in terms of element-wise $\ell_\infty$ norm; ensuring that the final estimator has the desired sparse structure.

### 3.1 Strong Statistical Guarantees of Closed-form Estimators

We now provide a statistical analysis of estimators in (8) under the following structural constraint:

**(C-Sparsity)** The "true" canonical exponential family parameter $\theta^*$ is exactly sparse with $k$ non-zero elements indexed by the support set $S$. All other elements in $S^c$ are zeros.

**Theorem 1.** *Consider any graphical model in* (1) *with sparse canonical parameter $\theta^*$ as stated in* (C-Sparsity). *Suppose we solve* (8) *setting the constraint bound $\lambda_n$ such that $\lambda_n \geq \left\|\theta^* - \mathcal{B}^*(\widehat{\phi})\right\|_\infty$.*

*(A) Then the optimal solution $\widehat{\theta}$ satisfies the following error bounds:*
$$\left\|\widehat{\theta} - \theta^*\right\|_\infty \leq 2\lambda_n \ , \quad \|\widehat{\theta} - \theta^*\|_2 \leq 4\sqrt{k}\lambda_n \ , \quad and \quad \left\|\widehat{\theta} - \theta^*\right\|_1 \leq 8k\lambda_n \ .$$

*(B) The support set of the estimate $\widehat{\theta}$ correctly excludes all true zero coordinates of $\theta^*$. Moreover, under the additional assumption that $\min_{s \in S} |\theta_s^*| \geq 3\left\|\theta^* - \mathcal{B}^*(\widehat{\phi})\right\|_\infty$, it correctly includes all non-zero coordinates of $\theta^*$.*

**Remarks.** Theorem 1 is a non-probabilistic result, and holds deterministically for any selection of $\lambda_n$ and any selection of $\mathcal{B}^*(\cdot)$. We would then use a probabilistic analysis when we applying the theorem to specific distributional settings and choices of the backward map $\mathcal{B}^*(\cdot)$.

We note that while the theorem analyses the case of sparsity structured parameters, our class of estimators as well as analyses can be seamlessly extended to more general structures (such as group sparsity and low rank), by substituting appropriate regularization functions in (8).

A key ingredient in our class of closed-form estimators is the proxy backward map $\mathcal{B}^*(\widehat{\phi})$. The conditions of the theorem require that this backward map has to be carefully constructed in order for the error bounds and sparsistency guarantees to hold. In the following sections, we will see how to precisely construct such backward maps $\mathcal{B}^*(\cdot)$ for specific problem instances, and then derive the corresponding consequences of our abstract theorem as corollaries.

## 4 Closed-form Estimators for Inverse Covariance Estimation in Gaussian Graphical Models

In this section, we derive a class of closed-form estimators for the multivariate Gaussian setting in Section 2.1. From our discussion of Gaussian graphical models in Section 2.1, the backward mapping from moments to the canonical parameters can be simply computed as $\mathcal{A}^*(\Sigma) = \Sigma^{-1}$, but only provided $\Sigma \in \mathcal{M}^o := \{\Sigma \in \mathbb{R}^{p \times p} : \Sigma \succ 0\}$. However, given the sample covariance, we cannot just compute the MLE as $\mathcal{A}^*(S) = S^{-1}$ since the sample covariance matrix is rank-deficient and hence does not belong the $\mathcal{M}^o$ under high-dimensional settings where $p > n$.

In our estimation framework (8), we thus use an alternative backward mapping $\mathcal{B}^*(\cdot)$ via a *thresholding* operator. Specifically, for any matrix $M \in \mathbb{R}^{p \times p}$, we consider the family of thresholding operators $T_\nu(M) : \mathbb{R}^{p \times p} \to \mathbb{R}^{p \times p}$ with thresholding parameter $\nu$, defined as $[T_\nu(M)]_{ij} := \rho_\nu(M_{ij})$ where $\rho_\nu(\cdot)$ is an element-wise thresholding operator. Soft-thresholding is a natural option, however, along the lines of [22], we can use arbitrary sparse thresholding operators satisfying the conditions:

**(C-Thresh)** For any input $a \in \mathbb{R}$, (i) $|\rho_\nu(a)| \leq |a|$, (ii) $|\rho_\nu(a)| = 0$ for $|a| \leq \nu$, and finally (iii) $|\rho_\nu(a) - a| \leq \nu$.

As long as $T_\nu(S)$ is *invertible* (which we shall examine in section 4.1), we can define $\mathcal{B}^*(S) := [T_\nu(S)]^{-1}$ and obtain the following class of estimators:

**Elem-GGM Estimation:** $\quad \underset{\Theta}{\text{minimize}}\ \|\Theta\|_{1,\text{off}}$ $\hfill (9)$

$$\text{s. t. } \left\|\Theta - [T_\nu(S)]^{-1}\right\|_{\infty,\text{off}} \le \lambda_n$$

where $\|\cdot\|_{\infty,\text{off}}$ is the off-diagonal element-wise $\ell_\infty$ norm as the dual of $\|\cdot\|_{1,\text{off}}$.

**Comparison with related work.** Note that [16] suggest a Dantzig-like estimator : $\text{minimize}_\Theta\ \|\Theta\|_1$ s. t. $\|S\Theta - I\|_\infty \le \lambda_n$ where both $\|\cdot\|_1$ and $\|\cdot\|_\infty$ are entry-wise ($\ell_1$ and $\ell_\infty$, respectively) norms for a matrix. This estimator applies penalty functions even for the diagonal elements so that the problem can be decoupled into multiple but much simpler optimization problems. It still requires solving $p$ linear programs with $2p$ linear constraints for each. On the other hand, the estimator from (9) has a closed-form solution as long as $T_\nu(S)$ is invertible.

## 4.1  Convergence Rates for Elem-GGM

In this section we derive a corollary of theorem 1 for Elem-GGM. A prerequisite is to show that $\mathcal{B}^*(S) := [T_\nu(S)]^{-1}$ is well-defined and "well-behaved". The following conditions define a broad class of Gaussian graphical models that satisfy this requirement.

**(C-MinInf$\Sigma$)** The true canonical parameter $\Theta^*$ of (3) has bounded induced operator norm such that $\|\Theta^*\|_\infty := \sup_{w \ne 0 \in \mathbb{R}^p} \frac{\|\Theta^* w\|_\infty}{\|w\|_\infty} \le \kappa_1$.

**(C-Sparse$\Sigma$)** The true covariance matrix $\Sigma^* := (\Theta^*)^{-1}$ is "approximately sparse" along the lines of Bickel and Levina [23]: for some positive constant $D$, $\Sigma_{ii}^* \le D$ for all diagonal entries, and moreover, for some $0 \le q < 1$ and $c_0(p)$, $\max_i \sum_{j=1}^p |\Sigma_{ij}^*|^q \le c_0(p)$. If $q = 0$, then this condition boils down to $\Sigma^*$ being sparse. We additionally require $\inf_{w \ne 0 \in \mathbb{R}^p} \frac{\|\Sigma^* w\|_\infty}{\|w\|_\infty} \ge \kappa_2$.

Now we are ready to utilize Theorem 1 and derive the convergence rates for our Elem-GGM (9).

**Corollary 1.** *Consider Gaussian graphical models* (3) *where the true parameter $\Theta^*$ has $k$ non-zero off-diagonal elements, and the conditions in* (C-MinInf$\Sigma$) *and* (C-Sparse$\Sigma$) *hold. Suppose that we solve the optimization problem in* (9) *with a generalized thresholding operator satisfying* (C-Thresh) *and setting $\nu := 16(\max_i \Sigma_{ii})\sqrt{\frac{10\tau \log p'}{n}} := a\sqrt{\frac{\log p'}{n}}$ for $p' := \max\{n, p\}$. Furthermore, suppose also that we select $\lambda_n := \frac{4\kappa_1 a}{\kappa_2}\sqrt{\frac{\log p'}{n}}$. Then, as long as $n > c_3 \log p'$, any optimal solution $\widehat{\Theta}$ of* (9) *satisfies*

$$\left\|\widehat{\Theta} - \Theta^*\right\|_{\infty,\text{off}} \le \frac{8\kappa_1 a}{\kappa_2}\sqrt{\frac{\log p'}{n}}\,, \quad \left\|\widehat{\Theta} - \Theta^*\right\|_{\text{F}} \le \frac{16\kappa_1 a}{\kappa_2}\sqrt{\frac{k \log p'}{n}}\,, \quad \left\|\widehat{\Theta} - \Theta^*\right\|_{1,\text{off}} \le \frac{32\kappa_1 a}{\kappa_2} k\sqrt{\frac{\log p'}{n}}$$

*with probability at least $1 - c_1 \exp(-c_2 \log p')$.*

We remark that the rates in Corollary 1 are asymptotically the same as those for standard $\ell_1$ regularized MLE estimators in (4); for instance, [13] show that $\|\widehat{\Theta}_{\text{MLE}} - \Theta^*\|_{\text{F}} = O\left(\sqrt{\frac{k \log p'}{n}}\right)$. This is remarkable given the simplicity of Elem-GGM.

# 5  Closed-form Estimators for Discrete Markov Random Fields

We now specialize our class of closed-form estimators (8) to the setting of discrete Markov random fields described in Section 2.1. In this case, computing the backward mapping $\mathcal{A}^*$ is non-trivial and typically intractable if the graphical structure has loops [18]. Therefore, we need an approximation of the backward map $\mathcal{A}^*$, for which we will leverage the tree-reweighted variational approximation discussed in Section 2.1. Consider the following map $\bar{\theta} := \mathcal{B}^*_{\text{trw}}(\widehat{\phi})$, where

$$\bar{\theta}_{s;j} = \log \widehat{\phi}_{s;j}, \text{ and } \quad \bar{\theta}_{st;jk} = \rho_{st} \log \frac{\widehat{\phi}_{st;jk}}{\widehat{\phi}_{s;j}\,\widehat{\phi}_{t;k}} \hfill (10)$$

where $\widehat{\phi}_{s;j} = \frac{1}{n}\sum_{i=1}^n \mathcal{I}[X_{s,i} = j]$ and $\widehat{\phi}_{st;jk} = \frac{1}{n}\sum_{i=1}^n \mathcal{I}[X_{s,i} = j]\mathcal{I}[X_{t,i} = k]$ are the empirical moments of the sufficient statistics in (6) (we define $0/0 := 1$). It was shown in [20] that $\mathcal{B}^*_{\text{trw}}(\cdot)$

satisfies the following property: the (pseudo)marginals computed by performing tree-reweighted variational inference with the parameters $\bar{\theta} := \mathcal{B}_{\mathrm{trw}}^*(\widehat{\phi})$ yield the sufficient statistics $\widehat{\phi}$. In other words, the approximate backward map $\mathcal{B}_{\mathrm{trw}}^*$ computes an element in the pre-image of the approximate forward map given by tree-reweighted variational inference. Since tree-reweighted variational inference approximates the true marginals well in practice, the map $\mathcal{B}_{\mathrm{trw}}^*(\cdot)$ is thus a great candidate for as an approximate backward map.

As an alternative to the $\ell_1$ regularized approximate MLE estimators (7), we thus obtain the following class of estimators using $\mathcal{B}_{\mathrm{trw}}^*(\cdot)$ as an instance of (8):

**Elem-DMRF Estimation:**   $\displaystyle\operatorname*{minimize}_{\theta} \|\theta\|_{1,E}$   (11)

$$\text{s.\,t. } \left\|\theta - \mathcal{B}_{\mathrm{trw}}^*(\widehat{\phi})\right\|_{\infty,E} \leq \lambda_n$$

where $\|\cdot\|_{\infty,E}$ is the maximum absolute value of edge-parameters as a dual of $\|\cdot\|_{1,E}$.

Note that given the empirical means of sufficient statistics, $\mathcal{B}_{\mathrm{trw}}^*(\widehat{\phi})$ can usually be obtained easily, *without* the need of explicitly specifying the log-partition function approximation $B(\cdot)$ in (7).

## 5.1 Convergence Rates for Elem-DRMF

We now derive the convergence rates of Elem-DRMF for the case where $\mathcal{B}^*(\cdot)$ is selected as in (10) following the tree reweighed approximation [20]. Let $\mu^*$ be the "true" marginals (or mean parameters) from the true log-partition function and true canonical parameter $\theta^*$: $\mu^* = \mathcal{A}(\theta^*)$. We shall require that the approximation $B_{\mathrm{trw}}(\cdot)$ be close enough to the true $A(\cdot)$ in terms of backward mapping. In addition we assume that true marginal distributions are strictly positive.

**(C-LogPartition)** $\left\|\theta^* - \mathcal{B}_{\mathrm{trw}}^*(\mu^*)\right\|_{\infty,E} \leq \epsilon$.

**(C-Marginal)** For all $s \in V$ and $j \in [m]$, the true singleton marginal $\mu_{s;j}^* := \mathbb{E}_{\theta^*}\big(\mathcal{I}[X_s = j]\big) = \mathbb{P}(X_s = j; \theta^*)$ satisfies $\epsilon_{\min} < \mu_{s;j}^*$ for some strictly positive constant $\epsilon_{\min} \in (0,1)$. Similarly, for all $s,t \in V$ and all $j,k \in [m]$, $\mu_{st;jk}^*$ satisfies $\epsilon_{\min} < \mu_{st;jk}^*$.

Now we are ready to utilize Theorem 1 to derive the convergence rates for our closed-form estimator (11) when $\theta^*$ has $k$ non-zero pairwise parameters $\theta_{st}^*$, where we recall the notatation that $\theta_{st} := \{\theta_{st;jk}\}_{j,k=1}^m$ is a collation of the edgewise parameters for edge $(s,t)$. We also define $\|\theta\|_{q,E} := \left(\sum_{s \neq t} \|\theta_{st}\|^q\right)^{1/q}$, for $q \in \{1,2,\infty\}$.

**Corollary 2.** *Consider discrete Markov random fields (6) when the true parameter $\theta^*$ has actually $k$ non-zero pair-wise parameters, and the conditions in* (C-LogPartition) *and* (C-Marginal) *also hold in these discrete MRFs. Suppose that we solve the optimization problem in (11) with $\mathcal{B}_{\mathrm{trw}}^*(\cdot)$ set as (10) using tree reweighed approximation. Furthermore, suppose also that we select $\lambda_n := \epsilon + c_1\sqrt{\frac{\log p}{n}}$ for some positive constant $c_1$ depending only on $\epsilon_{\min}$. Then, as long as $n > \frac{4c_1^2 \log p}{\epsilon_{\min}^2}$, there are universal positive constants $(c_2, c_3)$ such that any optimal solution $\widehat{\theta}$ of (11) satisfies*

$$\|\widehat{\theta} - \theta^*\|_{\infty,E} \leq 2\epsilon + 2c_1\sqrt{\frac{\log p}{n}}\,, \|\widehat{\theta} - \theta^*\|_{2,E} \leq 4\sqrt{k}\epsilon + 4c_1\sqrt{\frac{k \log p}{n}}\,, \|\widehat{\theta} - \theta^*\|_{1,E} \leq 8k\epsilon + 8c_1 k\sqrt{\frac{\log p}{n}}$$

*with probability at least $1 - c_2 \exp(-c_3 \log p')$.*

## 6 Experiments

In this section, we report a set of synthetic experiments corroborating our theoretical results on both Gaussian and discrete graphical models.

**Gaussian Graphical Models**   We now corroborate Corollary 1, and furthermore, compare our estimator with the $\ell_1$ regularized MLE in terms of **statistical performance** with respect to the parameter error $\|\widehat{\Theta} - \Theta^*\|_q$ for $q \in \{2,\infty\}$, as well as in terms of **computational performance**.

To generate true inverse covariance matrices $\Theta^*$ with a random sparsity structure, we follow the procedure described in [25, 24]. We first generate a sparse matrix $U$ whose non-zero entries are set to $\pm 1$ with equal probabilities. $\Theta^*$ is then set to $U^\top U$ and then a diagonal term is added to ensure

Table 1: Performance of our Elem-GM vs. state of the art QUIC algorithm [24] solving (4) under two different regimes: (Left) $(n, p) = (800, 1600)$, (Right) $(n, p) = (5000, 10000)$.

| | K | Time(sec) | $\ell_F$ (off) | $\ell_\infty$ (off) | FPR | TPR |
|---|---|---|---|---|---|---|
| Elem-GM | 0.01 | < 1 | 6.36 | 0.1616 | 0.48 | 0.99 |
| | 0.02 | < 1 | 6.19 | 0.1880 | 0.24 | 0.99 |
| | 0.05 | < 1 | 5.91 | 0.1655 | 0.06 | 0.99 |
| | 0.1 | < 1 | 6 | 0.1703 | 0.01 | 0.97 |
| QUIC | 0.5 | 2575.5 | 12.74 | 0.11 | 0.52 | 1.00 |
| | 1 | 1009 | 7.30 | 0.13 | 0.35 | 0.99 |
| | 2 | 272.1 | 6.33 | 0.18 | 0.16 | 0.99 |
| | 3 | 78.1 | 6.97 | 0.21 | 0.07 | 0.94 |
| | 4 | 28.7 | 7.68 | 0.23 | 0.02 | 0.86 |

| | K | Time(sec) | $\ell_F$ (off) | $\ell_\infty$ (off) | FPR | TPR |
|---|---|---|---|---|---|---|
| Elem-GM | 0.05 | 47.3 | 11.73 | 0.1501 | 0.13 | 1.00 |
| | 0.1 | 46.3 | 8.91 | 0.1479 | 0.03 | 1.00 |
| | 0.5 | 45.8 | 5.66 | 0.1308 | 0.0 | 1.00 |
| | 1 | 46.2 | 8.63 | 0.1111 | 0.0 | 0.99 |
| QUIC | 2 | * | * | * | * | * |
| | 2.5 | * | * | * | * | * |
| | 3 | $4.8 \times 10^4$ | 9.85 | 0.1083 | 0.06 | 1.00 |
| | 3.5 | $2.7 \times 10^4$ | 10.51 | 0.1111 | 0.04 | 0.99 |

Table 2: Performance of Elem-DMRF vs. the regularized MLE-based approach of [12] for structure recovery of DRMFs.

| Graph Type | # Parameters | Method | Time(sec) | TPR | FNR |
|---|---|---|---|---|---|
| Chain Graph | 128 | Elem-DMRF | 0.17 | 0.87 | 0.01 |
| | | Regularized MLE | 7.30 | 0.81 | 0.01 |
| | 2000 | Elem-DMRF | 21.67 | 0.79 | 0.12 |
| | | Regularized MLE | 4315.10 | 0.75 | 0.21 |
| Grid Graph | 128 | Elem-DMRF | 0.17 | 0.97 | 0.01 |
| | | Regularized MLE | 7.99 | 0.84 | 0.02 |
| | 2000 | Elem-DMRF | 21.68 | 0.80 | 0.12 |
| | | Regularized MLE | 4454.44 | 0.77 | 0.18 |

$\Theta^*$ is positive definite. Finally, we normalize $\Theta^*$ with $\max_{i=1}^p \Theta_{ii}^*$ so that the maximum diagonal entry is equal to 1. We control the number of non-zeros in $U$ so that the number of non-zeros in the final $\Theta^*$ is approximately $10p$. We additionally set the number of samples $n$ to half of the number of variables $p$. Note that though the number of variables is $p$, the total number of entries in the canonical parameter consisting of the covariance matrix is $O(p^2)$.

Table 1 summarizes the performance of our closed-form estimators in terms of computation time, $\|\widehat{\Theta} - \Theta^*\|_{\infty,\text{off}}$ and $\|\widehat{\Theta} - \Theta^*\|_{\text{F,off}}$. We fix the thresholding parameter $\nu = 2.5\sqrt{\log p/n}$ for all settings, and vary the regularization parameter $\lambda_n = K\sqrt{\log p/n}$ to investigate how this regularizer affects the final estimators. Baselines are $\ell_1$ regularized MLE estimators in (4); we use QUIC algorithms [24], which is one of the fastest way to solve (4). In the table, we show the results of the QUIC algorithm run with a tolerance $\epsilon = 10^{-4}$; * indicates that the algorithm does not stop within 15 hours. In Appendix, we provide more extensive comparisons including receiver operator curves (ROC) for these methods for settings in Table 1. As can be seen from the table and the figure, the performance of Elem-GM estimators is both **statistically** competitive in terms of all types of errors and support set recovery, while performing much better **computationally** than classical methods based on $\ell_1$ regularized MLE.

**Discrete Graphical Models** We consider two different classes of pairwise graphical models: chain graphs and grids. For each case, the size of the alphabet is set to $m = 3$; the true parameter vector $\theta^*$ is generated by sampling each non-zero entry from $N(0, 1)$.

We compare Elem-DMRF with the group-sparse regularized MLE-based approach of Jalali et al. [12], which uses group $\ell_1/\ell_2$ regularization, where all the parameters of an edge form a group, so as to encourage sparsity in terms of the edges, and which we solved using proximal gradient descent. While our estimator in (11) used vanilla sparsity, we used a simple extension to the group-sparse structured setting; please see Appendix E for more details. For both methods, the tuning parameter is set to $\lambda_n = c\sqrt{\log p/n}$, where $c$ is selected using cross-validation. We use 20 simulation runs where for each run $n = p/2$ samples are drawn from the distribution specified by $\theta^*$.

We report true positive rates, false positive rates and timing for running each method. We note that the timing is for running each method without counting the time spent in the cross-validation process (Had we taken the cross-validation into account, the advantage of our method would be even more pronounced, since the entire path of solutions can be computed via simple group-wise thresholding operations.) The results in Table 2 show that Elem-DMRF is much faster than its MLE-based counterpart, and yield competitive results in terms of structure recovery.

**Acknowledgments** E.Y and P.R. acknowledge the support of ARO via W911NF-12-1-0390 and NSF via IIS-1149803, IIS-1320894, IIS-1447574, and DMS-1264033

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
