[Supplementary Material · Elem_exp_final_sup.pdf]



Figure 1: Receiver operator curves for support set recovery task when $(n, p) = (800, 1600)$ (Left), $(n, p) = (5000, 10000)$ (Right).

## Appendix

## A    Proof of Theorem 1

Let $\Delta$ be the error vector, $\widehat{\theta} - \theta^*$. Since we choose $\lambda_n$ greater than $\|\theta^* - [\nabla B]^{-1}(\widehat{\phi})\|_\infty$,

$$\|\Delta\|_\infty = \|\widehat{\theta} - [\nabla B]^{-1}(\widehat{\phi}) + [\nabla B]^{-1}(\widehat{\phi}) - \theta^*\|_\infty$$
$$\leq \|\widehat{\theta} - [\nabla B]^{-1}(\widehat{\phi})\|_\infty + \|\theta^* - [\nabla B]^{-1}(\widehat{\phi})\|_\infty \leq 2\lambda_n. \tag{12}$$

At the same time, by the fact that $\theta^*_{S^c} = \mathbf{0}$, and the decomposability of $\|\cdot\|_1$ with respect to $(S, S^c)$,

$$\|\theta^*\|_1 = \|\theta^*\|_1 + \|\Delta_{S^c}\|_1 - \|\Delta_{S^c}\|_1$$
$$= \|\theta^* + \Delta_{S^c}\|_1 - \|\Delta_{S^c}\|_1$$
$$\overset{(i)}{\leq} \|\theta^* + \Delta_{S^c} + \Delta_S\|_1 + \|\Delta_S\|_1 - \|\Delta_{S^c}\|_1$$
$$= \|\theta^* + \Delta\|_1 + \|\Delta_S\|_1 - \|\Delta_{S^c}\|_1 \tag{13}$$

where the equality $(i)$ holds by the triangle inequality of $\ell_1$ norm. Now, since we minimize the objective $\|\theta\|_1$ in (8), we obtain the inequality of $\|\theta^* + \Delta\|_1 = \|\widehat{\theta}\|_1 \leq \|\theta^*\|_1$. Combining this inequality with (13), we have

$$0 \leq \|\Delta_S\|_1 - \|\Delta_{S^c}\|_1. \tag{14}$$

Armed with inequalities (12) and (14), we utilize the Hölder's inequality and the decomposability of $\|\cdot\|_1$ in order to compute the error bound:

$$\|\Delta\|_2^2 = \langle \Delta, \Delta \rangle \leq \|\Delta\|_\infty \|\Delta\|_1 \leq \|\Delta\|_\infty \left( \|\Delta_S\|_1 + \|\Delta_{S^c}\|_1 \right). \tag{15}$$

Since the error vector $\Delta$ satisfies the property: $\|\Delta_{S^c}\|_1 \leq \|\Delta_S\|_1$ from (14),

$$\|\Delta\|_2^2 \leq 2\|\Delta\|_\infty \|\Delta_S\|_1. \tag{16}$$

Combining all the pieces together yields

$$\|\Delta\|_2^2 \leq 4\lambda_n \sqrt{k} \|\Delta_S\|_2. \tag{17}$$

Notice that the projection operator is non-expansive, $\|\Delta_S\|_2^2 \leq \|\Delta\|_2^2$. Hence, we obtain $\|\Delta_S\|_2 \leq 4\lambda_n\sqrt{k}$, and plugging it back into (17) yields the error bound, $\|\widehat{\theta} - \theta^*\|_2$.

Finally, the error bound in terms of $\ell_1$, is straightforward from the following reasoning:

$$\|\Delta\|_1 = \|\Delta_S\|_1 + \|\Delta_{S^c}\|_1 \leq 2\|\Delta_S\|_1 \leq 2\sqrt{k}\|\Delta_S\|_2 \leq 8\lambda_n k.$$

# B  Useful lemma(s)

**Lemma 1** (Theorem 1 of [22, 23]). *Let $\delta$ be $\max_{ij}\left|\left[\frac{X^\top X}{n}\right]_{ij} - \Sigma_{ij}\right|$. Suppose that $\nu \geq 2\delta$. Then, under the conditions* (C-Thresh) *and* (C-Sparse$\Sigma$)*, we can deterministically guarantee that the spectral norm of error is bounded as follows*

$$\left\|\!\left\|T_\nu(S) - \Sigma\right\|\!\right\|_\infty \leq 5\nu^{1-q}c_0(p) + 3\nu^{-q}c_0(p)\delta. \tag{18}$$

**Lemma 2** (Lemma 1 of [13]). *Let $\mathcal{A}$ be the event that*

$$\left\|\frac{X^\top X}{n} - \Sigma\right\|_\infty \leq 8(\max_i \Sigma_{ii})\sqrt{\frac{10\tau \log p'}{n}}$$

*where $p' := \max\{n, p\}$ and $\tau$ is any constant greater than 2. Suppose that the design matrix $X$ is i.i.d. sampled from $\Sigma$-Gaussian ensemble with $n \geq 40 \max_i \Sigma_{ii}$. Then, the probability of event $\mathcal{A}$ occurring is at least $1 - 4/p'^{\tau-2}$.*

**Lemma 3** (Lemma 3 of [19]). *For discrete graphical models in* (6)*,*

$$\|\phi - \mu^*\|_\infty \leq 2\sqrt{\frac{\log p}{n}}$$

*with probability at least $1 - 2\exp(-2\log p)$.*

# C  Proof of Corollary 1

In order to utilize Theorem 1 for this specific case, we only need to show that $\|\Theta^* - [T_\nu(S)]^{-1}\|_{\infty,\text{off}} \leq \lambda_n$ for the setting of $\lambda_n$ in the statement:

$$\begin{aligned}
\left\|\Theta^* - [T_\nu(S)]^{-1}\right\|_{\infty,\text{off}} &= \left\|[T_\nu(S)]^{-1}\big(T_\nu(S)\Theta^* - I\big)\right\|_{\infty,\text{off}} \\
&\leq \left\|\!\left\|[T_\nu(S)]^{-1}\right\|\!\right\|_\infty \left\|T_\nu(S)\Theta^* - I\right\|_{\infty,\text{off}} = \left\|\!\left\|[T_\nu(S)]^{-1}\right\|\!\right\|_\infty \left\|\Theta^*\big(T_\nu(S) - \Sigma^*\big)\right\|_{\infty,\text{off}} \\
&\leq \left\|\!\left\|[T_\nu(S)]^{-1}\right\|\!\right\|_\infty \left\|\!\left\|\Theta^*\right\|\!\right\|_\infty \left\|T_\nu(S) - \Sigma^*\right\|_{\infty,\text{off}}. 
\end{aligned} \tag{19}$$

We first compute the upper bound of $\left\|\!\left\|[T_\nu(S)]^{-1}\right\|\!\right\|_\infty$. By the selection $\nu$ in the statement, Lemma 1 and 2 hold with probability at least $1 - 4/p'^{\tau-2}$. Armed with (18), we use the triangle inequality of norm and the condition (C-Sparse$\Sigma$): for any $w$

$$\left\|T_\nu(S)w\right\|_\infty = \left\|T_\nu(S)w - \Sigma w + \Sigma w\right\|_\infty \geq \left\|\Sigma w\right\|_\infty - \left\|\big(T_\nu(S) - \Sigma\big)w\right\|_\infty$$

$$\overset{(i)}{\geq} \kappa_2 \|w\|_\infty - \left\|\big(T_\nu(S) - \Sigma\big)w\right\|_\infty \geq \Big(\kappa_2 - \left\|\!\left\|T_\nu(S) - \Sigma\right\|\!\right\|_\infty\Big)\|w\|_\infty$$

where the inequality (i) uses the condition (C-Sparse$\Sigma$). Now, by Lemma 1 with the selection of $\nu$, we have

$$\left\|\!\left\|T_\nu(S) - \Sigma\right\|\!\right\|_\infty \leq c_1\Big(\frac{\log p'}{n}\Big)^{(1-q)/2}c_0(p)$$

where $c_1$ is a constant related only on $\tau$ and $\max_i \Sigma_{ii}$. Specifically, it is defined as $6.5\big(16(\max_i \Sigma_{ii})\sqrt{10\tau}\big)^{1-q}$. Hence, as long as $n > \big(\frac{2c_1 c_0(p)}{\kappa_2}\big)^{\frac{2}{1-q}}\log p'$ as stated, so that $\|T_\nu(S) - \Sigma\|_\infty \leq \frac{\kappa_2}{2}$, we can conclude that $\left\|T_\nu(S)w\right\|_\infty \geq \frac{\kappa_2}{2}\|w\|_\infty$, which implies $\left\|\!\left\|[T_\nu(S)]^{-1}\right\|\!\right\|_\infty \leq \frac{2}{\kappa_2}$.

The remaining term in (19) is $\|T_\nu(S) - \Sigma^*\|_{\infty,\text{off}}$ ; $\|T_\nu(S) - \Sigma^*\|_{\infty,\text{off}} \leq \|T_\nu(S) - S\|_{\infty,\text{off}} + \|S - \Sigma^*\|_{\infty,\text{off}}$. By construction of $T_\nu(\cdot)$ in (C-Thresh) and by Lemma 2, we can confirm that $\|T_\nu(S) - S\|_{\infty,\text{off}}$ as well as $\|S - \Sigma^*\|_{\infty,\text{off}}$ can be upper-bounded by $\nu$.

By combining all together, we can confirm that the selection of $\lambda_n$ satisfies the requirement of Theorem 1, which completes the proof.

## D  Proof of Corollary 2

As in proof of Corollary 1, we need to show that $\|\theta^* - \mathcal{B}^*_{\mathrm{trw}}(\widehat{\phi})\|_{\infty,E} \leq \lambda_n$ for the setting of $\lambda_n$ in the statement:

$$\left\|\theta^* - \mathcal{B}^*_{\mathrm{trw}}(\widehat{\phi})\right\|_{\infty,E}$$
$$= \left\|\theta^* - \mathcal{B}^*_{\mathrm{trw}}(\mu^*) + \mathcal{B}^*_{\mathrm{trw}}(\mu^*) - \mathcal{B}^*_{\mathrm{trw}}(\widehat{\phi})\right\|_{\infty,E}$$
$$\leq \left\|[\theta^* - \mathcal{B}^*_{\mathrm{trw}}(\mu^*)]\right\|_{\infty,E} + \left\|[\mathcal{B}^*_{\mathrm{trw}}(\mu^*) - \mathcal{B}^*_{\mathrm{trw}}(\widehat{\phi})]\right\|_{\infty,E}$$
$$\leq \epsilon + \left\|\mathcal{B}^*_{\mathrm{trw}}(\mu^*) - \mathcal{B}^*_{\mathrm{trw}}(\widehat{\phi})\right\|_{\infty,E}$$

Now, let us focus on the second term above, where $\mathcal{B}^*_{\mathrm{trw}}(\cdot)$ is defined in (10). For all any combination of $(st; jk)$, we have

$$\left| \rho_{st} \log \frac{\mu^*_{st;jk}}{\mu^*_{s;j}\,\mu^*_{t;k}} - \rho_{st} \log \frac{\widehat{\phi}_{st;jk}}{\widehat{\phi}_{s;j}\,\widehat{\phi}_{t;k}} \right| \leq \left| \log \frac{\mu^*_{st;jk}}{\mu^*_{s;j}\,\mu^*_{t;k}} - \log \frac{\widehat{\phi}_{st;jk}}{\widehat{\phi}_{s;j}\,\widehat{\phi}_{t;k}} \right|$$
$$= \left| \left(\log \mu^*_{st;jk} - \log \widehat{\phi}_{st;jk}\right) + \left(\log \widehat{\phi}_{s;j} - \log \mu^*_{s;j}\right) + \left(\log \widehat{\phi}_{t;k} - \log \mu^*_{t;k}\right) \right|$$
$$\leq \left| \log \mu^*_{st;jk} - \log \widehat{\phi}_{st;jk} \right| + \left| \log \widehat{\phi}_{s;j} - \log \mu^*_{s;j} \right| + \left| \log \widehat{\phi}_{t;k} - \log \mu^*_{t;k} \right|$$

By Lemma 3, $\|\phi - \mu^*\|_\infty \leq c_1 \sqrt{\frac{\log p}{n}}$ with at least probability $1 - 2\exp(-2\log p)$. Therefore, for any index $\alpha$, we have

$$\left| \log \widehat{\phi}_\alpha - \log \mu^*_\alpha \right| = \log \frac{\max\{\widehat{\phi}_\alpha, \mu^*_\alpha\}}{\min\{\widehat{\phi}_\alpha, \mu^*_\alpha\}} \leq \log \left( \frac{\min\{\widehat{\phi}_\alpha, \mu^*_\alpha\} + c_1\sqrt{\frac{\log p}{n}}}{\min\{\widehat{\phi}_\alpha, \mu^*_\alpha\}} \right)$$

$$\leq \log \left( 1 + \frac{c_1}{\min\{\widehat{\phi}_\alpha, \mu^*_\alpha\}} \sqrt{\frac{\log p}{n}} \right) \leq \frac{c_1}{\min\{\widehat{\phi}_\alpha, \mu^*_\alpha\}} \sqrt{\frac{\log p}{n}}.$$

If $n > \frac{4c_1^2 \log p}{\epsilon_{\min}^2}$, then $\widehat{\phi}_\alpha \geq \mu^*_\alpha - c_1\sqrt{\frac{\log p}{n}} \geq \mu^*_\alpha - \frac{\epsilon_{\min}}{2} \geq \frac{\epsilon_{\min}}{2}$ again by Lemma 3 and (C-Marginal). Hence, we can conclude $\left| \log \widehat{\phi}_\alpha - \log \mu^*_\alpha \right| \leq \frac{2c_1}{\epsilon_{\min}} \sqrt{\frac{\log p}{n}}$, and finally we have $\|\theta^* - \mathcal{B}^*_{\mathrm{trw}}(\widehat{\phi})\|_{\infty,E} \leq \frac{6c_1}{\epsilon_{\min}} \sqrt{\frac{\log p}{n}}$.

## E  Extension to Group Sparsity in DMRFs

A pertinent structural constraint for DMRFs is that of group-sparsity, where all the parameters of an edge are grouped together, so as to encourage sparsity in terms of the edges. Specifically, for each pair of nodes $(s,t)$ in the DMRF, denote by $G_{s,t}$ the group of indices corresponding to the parameter group $\{\theta_{s,t;j,k} : j, k \in [m]\}$. Let $\theta_{G_{s,t}}$ denote the corresponding parameter sub-vector. Let $\mathcal{G} := \{G_{s,t} : s, t \in V\}$. A natural regularization function for such a setting is the following group-structured $\ell_1/\ell_\alpha$ norm defined as $\|\theta\|_{\mathcal{G},\alpha,E} := \sum_{(s,t)\in V} \|\theta_{G_{s,t}}\|_\alpha$, where $\alpha$ is a constant between 2 and $\infty$.

We then consider the following variant of Elem-DMRF, with the regularization function set to the above group-structured norm:

$$\underset{\theta}{\text{minimize}} \ \|\theta\|_{\mathcal{G},\alpha,E}$$
$$\text{s.t.} \ \left\|\theta - \mathcal{B}^*_{\mathrm{trw}}(\widehat{\phi})\right\|^*_{\mathcal{G},\alpha,E} \leq \lambda_n$$

where $\|\theta\|^*_{\mathcal{G},\alpha,E} := \max_{(s,t)} \|\theta_{G_{s,t}}\|_{\alpha^*}$ for a constant $\alpha^*$ satisfying $\frac{1}{\alpha} + \frac{1}{\alpha^*} = 1$.

It can easily be seen that the estimator is still available in closed-form via group-wise soft-thresholding of $\mathcal{B}^*_{\mathrm{trw}}(\widehat{\phi})$. We note that our theoretical analysis can be naturally extended to such group sparsity structure (and to other structures such as low rank). We will consider doing so in future work.