[Reviews · NeurIPS 2014]

Submitted by Assigned_Reviewer_18

This paper proposes closed-form estimators for sparsity-structured graphical models, expressed as exponential family distributions, under high-dimensional settings.
The paper is strong in both the methodology and theory aspect.
The paper is well structured and written.
The paper would be even better if the authors can discuss the connection between the method in the paper with previous methods which can provide close-form estimators for special MRFs such as tree-width=1 or planar MRF.
Summary: A good paper

Submitted by Assigned_Reviewer_25

The authors provide new closed-form methods for parameter estimation in graphical models. They extend a new line of work on elementary estimators to the realm of Gaussian and discrete graphical models, and derive theoretical guarantees for the estimators proposed in their paper.

The approach supplied by the authors is quite fascinating in its simplicity. Indeed, parameter estimation in Gaussian and discrete graphical models becomes a computationally challenging task for large p and n, and the closed-form estimators may lead to massive speed-ups. The method has some drawbacks, however: One chief drawback for Elem-GGM is the seemingly stringent assumption C-Sparse Sigma, which enforces (approximate) sparsity on the covariance matrix, in addition to the assumed sparsity on the inverse covariance matrix. The reviewer wonders whether there are settings where both types of sparsity would hold simultaneously. In the experiments, the authors do not seem to be concerned about whether Sigma satisfies the assumption, and only take care to make Sigma^{-1} sparse.

Theoretical details for closed-form estimators of DRMFs are also lacking. The main problem is that there is no way of knowing when assumption C-LogPartition should hold; even in the literature on tree-reweighted variational approximations, theoretical guarantees on the closeness of the approximation to the true parameters are scarce. The authors assume that the output of (11) will be sufficiently accurate, whereas its accuracy will vary greatly depending on the true parameters, choice of rho, and graph structure.
Summary: The content of the paper is interesting, but the assumptions involved in validating the theory seem too strong for the scenarios under consideration.

Submitted by Assigned_Reviewer_30

Focusing on high dimensionality, this paper proposes a class of closed form of estimators for graphical models in exponential family distribution with sparsity (particularly regularized by L1 norm) structures. Using backward map and thresholding operators, the resulted estimators are simple to implement and efficient.

The paper is overall clear. The good experimental results and simplicity of implementation suggest the usefulness of the proposed method. The supplementary contains enough details.

The originality is middle, considering it extends a step further from the recent ICML'2014 paper "Elementary Estimators for High-Dimensional Linear Regression".

The work is technically correct, and the results are promising on both efficiency and effectiveness.
Summary: This work is interesting and works well empirically. With carefully designed backward maps, this framework could be applied on models within a wide range.

Submitted by Assigned_Reviewer_45

I personally view this paper quite positively. I feel that the paper was well written, assumptions and theorems clearly stated, and (interestingly) the proofs quite simple and well written. On reading the reviewer discussion and author feedback, there seemed to be two main issues arising

(1) Too similar to earlier papers? I think not - even if the high-level idea is similar, the devil is in the details, which need to be fleshed out to get the right rates of convergence, good performance in the experiments, and so on. If I gave someone capable the earlier paper and asked them to derive the kinds of results seen in this one, it would not be "straightforward" to extend those results - it really would need some hard work.

(2) Too harsh assumptions? This might be true but does not warrant rejection - the idea of closed-formed estimators for these problems is *important* and the community deserves to be aware of these advances, especially because they have been shown to work very well practically in this paper - it really is a game-changer compared to graph lasso or even neighborhood estimation which needs lasso-solvers. I do think that further research will lighten the assumptions used (or get a better understanding of them).

The experiments are very promising, this stuff actually seems to work fast and well, and so it is possibly a case of a gap between what works and what's provable. I suspect future theory developed by other people will weaken these assumptions - people will want to work on this or extend it etc because it seems to work well.

In summary, I clearly stand on the side of acceptance.
Summary: While I would have been more skeptical for other papers, I feel the ideas behind this paper have the potential to be disruptive in how people use these sparse estimators in practice - there is huge scope for application and theory in the future and that makes me believe that this paper should be out there for people to read and discuss today and not 6 months from now.
Author Feedback
Author rebuttal: We thank the reviewers for their comments and feedback.

Here are the two main contributions of our submission, each being individually meaningful:

1. From an algorithmic viewpoint, we provide computationally efficient closed-form estimators for graphical models. This is surprising because till recently these were solved using iterative greedy procedures, and recently via iterative algorithms for regularized convex programs. We present experimental results for the most natural settings, and interestingly, our estimators outperform compared methods both computationally and statistically.

2. From an analytical standpoint, we analyze the statistical guarantees of our closed-form estimators for graphical models.

I. Distinction with the 2014 ICML paper "Elementary Estimators for High-Dimensional Linear Regression":

The 2014 ICML paper above provided closed-form estimators for high-dimensional linear regression. Our work shares only its high-level motivations with that paper — on deriving statistically consistent closed-form estimators. However, to do so for the graphical model learning problem is considerably more complex (indeed: for linear regression under low-dimensional settings, the ordinary least squares estimator is already in closed-form). Moreover, the need of closed-form estimators for graphical models such as discrete or Gaussian graphical models is even more pressing, because, in contrast to linear regression, standard iterative methods for these are very expensive and can only handle limited-sized problems (and accordingly, are an active area of research in ML and Statistics).

II. Condition for the Gaussian graphical model case: Our corollary requires the condition that every row of \Sigma has bounded \ell_q norm (when q is between 0 and 1), which allows \Sigma that is not truly sparse but has many small entries. Moreover, this is just a sufficient condition; at the cost of presentational simplicity, our theorem can be shown to hold under more general conditions, which would be satisfied for instance even by merely diagonally dominant \Sigma. These conditions actually include many natural cases, for instance, the case where \Theta is chain, star, 4NN or 8NN lattice structures. We will clarify this in the final version.

It remains as a future work to find *necessary* conditions under which our closed-form estimators are consistent; our closed-form estimators might require possibly stronger condition than that required in standard \ell_1 regularized MLE-like estimators, but provide huge computation gains. Nevertheless, statistical guarantees still hold for a quite broad class of \Theta, and we can experience of huge improvement for such \Theta, as we showed in our theorem and experiments (In the experiment, we consider the most difficult and the most natural case: randomly generated sparse graph \Theta).

Assigned_Reviewer_25:

We would like to correct the reviewer’s misunderstanding of our assumption; Corollary 2 *does not* require eq (11) is accurate. (C-LogPartition) is about the accuracy of approximated log-partition function. (C-LogPartition) requires approximated backward mapping of the *population* mean \mu^* is close to \theta^* with \eps error. If the approximate backward mapping is exact, then (C-LogPartition) trivially holds with \eps=0. Eq (11) can be large even though the approximation is exact.

This \eps is still dependent on the variational approximations for discrete graphical models as the reviewer pointed out. Nevertheless, Corollary 2 is useful: it is completely complementary to advances in providing guarantees for variational approximations (such as tree-reweighted approx.); note that such guarantees are outside the scope of this paper, but are an active area of current research for varied specialized graphical models.